# Rethinking the Explanation of Graph Neural Network via Non-parametric Subgraph Matching

## Abstract

The great success in *graph neural networks (GNNs)* provokes the question about explainability: "Which fraction of the input graph is the most determinant to the prediction?" However, current approaches usually resort to a black-box to decipher another black-box (i.e., GNN), making it difficult to understand how the explanation is made. Based on the observation that graphs typically share some joint motif patterns, we propose a novel subgraph matching framework named MatchExplainer to explore explanatory subgraphs. It couples the target graph with other counterpart instances and identifies the most crucial joint substructure by minimizing the node corresponding-based distance between them. After that, an external graph ranking is followed to select the most informative substructure from all subgraph candidates. Thus, MatchExplainer is entirely non-parametric. Moreover, present graph sampling or node dropping methods usually suffer from the false positive sampling problem. To ameliorate that issue, we take advantage of MatchExplainer to fix the most informative portion of the graph and merely operate graph augmentations on the rest less informative part, which is dubbed as MatchDrop. We conduct extensive experiments on both synthetic and real-world datasets, showing the effectiveness of our MatchExplainer by outperforming all parametric baselines with large margins. Additional results also demonstrate that our MatchDrop is a general paradigm to be equipped with GNNs for enhanced performance.

## 1 Introduction

*Graph neural networks (GNNs)* have drawn broad interest due to their success for learning representations of graph-structured data, such as social networks (Fan et al., 2019), knowledge graphs (Schlichtkrull et al., 2018), traffic networks (Geng et al., 2019), and microbiological graphs (Gilmer et al., 2017). Despite their remarkable efficacy, GNNs lack transparency as the rationales of their predictions are not easy for humans to comprehend. This prohibits practitioners from not only gaining an understanding of the network characteristics, but correcting systematic patterns of mistakes made by models before deploying them in real-world applications.

Recently, extensive efforts have been devoted to studying the explainability of GNNs (Yuan et al., 2020). Researchers strive to answer the questions like "What knowledge of the input graph is the most dominantly important in the model's decision?" Towards this end, feature attribution and selection (Selvaraju et al., 2017; Sundararajan et al., 2017; Ancona et al., 2017) is a prevalent paradigm. They distribute the model's outcome prediction to the input graph via gradient-like signals (Baldassarre & Azizpour, 2019; Pope et al., 2019; Schnake et al., 2020), mask or attention scores (Ying et al., 2019; Luo et al., 2020), or prediction changes on perturbed features (Schwab & Karlen, 2019; Yuan et al., 2021), and then choose a salient substructure as the explanation.

Nonetheless, the latest approaches are all deep learning-based and rely on a network to parameterize the generation process of explanations (Vu & Thai, 2020; Wang et al., 2021b). We argue that depending on another black-box to comprehend the prediction of the target black-box (i.e., GNNs) is sub-optimal, since the behavior of those explainers is hard to interpret. These black-boxes, indeed, always fail to give a clue of how they find proper explanatory subgraphs. In contrast, a decent

explainer ought to provide clear insights of how it captures and values this substructure. Otherwise, a lack of interpretability in explainers can undermine our trust in them. Moreover, some prior works (Chen et al., 2018; Ying et al., 2019; Yuan et al., 2021) independently excavate explanations for each instance without explicitly referring to other training data in the inference phase. They ignore the fact that different essential subgraph patterns are shared by different groups of graphs, which can be the key to decipher the decision of GNNs. These frequently occurred motifs usually contain rich semantic meanings and indicate the characteristics of the whole graph instance (Henderson et al., 2012; Zhang et al., 2020; Banjade et al., 2021). For example, the hydroxide group (-OH) in small molecules typically results in higher water solubility, and the pivotal role of functional groups has also been proven in protein structure prediction (Senior et al., 2020).

To overcome these drawbacks, we propose to mine the explanatory motif in a subgraph matching manner. In contrast to a learnable network, we design a non-parametric algorithm dubbed MatchExplainer with no need for training, which is composed of two stages. At the first stage, it marries the target graph iteratively with other counterpart graphs and endeavors to explore the most crucial joint substructure by minimizing the node corresponding-based distance in the high-dimensional feature space. Since the counterpart graphs are diverse, the explanations at the first stage of MatchExplainer can be non-unique for the same instance. Thus, an external graph ranking technique is followed as the second stage of MatchExplainer to pick out the most appropriate one. To be explicit, it examines the important role that these substructures plays in determining the graph property by subtracting the subgraphs from the original input graph and testing the prediction of the remaining part.

Our MatchExplainer not only shows great potential in fast discovering the explanations for GNNs, but also can be employed to enhance the traditional graph augmentation methods. Though exhibiting strong power in preventing over-fitting and over-smoothing, present graph sampling or node dropping mechanisms suffer from the false positive sampling problem. That is, nodes or edges of the most informative substructure are accidentally dropped or erased but the model is still required to forecast the original property, which can be misleading. To alleviate this obstacle, we take advantage of MatchExplainer and introduce a simple technique called MatchDrop. Specifically, it first digs out the explanatory subgraph by means of MatchExplainer and keeps this part unchanged. Then the graph sampling or node dropping is implemented solely on the remaining less informative part. As a consequence, the core fraction of the input graph that reveals the label information is not affected and the false positive sampling issue is effectively mitigated.

To summarize, we are the foremost to investigate the explainability of GNNs from the perspective of non-parametric subgraph matching to the best of our knowledge. Extensive experiments on synthetic and real-world applications demonstrate that our MatchExplainer can find the explanatory subgraphs fast and accurately with state-of-the-art performance. Apart from that, we empirically show that our MatchDrop can serve as an efficient way to promote the graph augmentation methods.

## 2 PRELIMINARY AND TASK DESCRIPTION

In this section, we begin with the description of the GNN explanation task and then briefly review the relevant background of graph matching and mutual information theory.

**Explanations for GNNs.** Let $f_Y$ denote a well-trained GNN to be explained, which gives the prediction $\hat{y}_{\mathcal{G}}$ of the input graph $\mathcal{G}$ to approximate the ground-truth label $y_{\mathcal{G}}$. Without loss of generality, we consider the problem of explaining a graph classification task (Ying et al., 2019; Yuan et al., 2020) as to find an explainer $f_S$ that discovers the subgraph $\mathcal{G}_S$ from input graph $\mathcal{G}$ by:

$$\underset{f_S}{\arg\min}\, \mathcal{R}(f_Y \circ f_S(\mathcal{G}), \hat{y}_{\mathcal{G}}), \text{s.t.} |f_S(\mathcal{G})| \leq K, \tag{1}$$

where $\mathcal{R}(\cdot, \cdot)$ is the risk function, which is usually implemented as a cross-entropy (CE) loss or a mean squared error (MSE) loss, and $|\cdot|$ returns the graph size (namely the number of nodes in this paper), and $K$ is a prefixed constraint.

**Graph Matching.** As a classic combinatorial problem, graph matching is known to be NP-hard (Loiola et al., 2007). Addressing it requires expensive, complex, and impractical solvers, hence plenty of inexact but practical solutions (Wang et al., 2020) have been proposed. Given two different

graphs $\mathcal{G}_1 = (\mathcal{V}_1, \mathcal{E}_1)$ and $\mathcal{G}_2 = (\mathcal{V}_2, \mathcal{E}_2)$ with $N_1$ and $N_2$ nodes respectively, the matching between them can be generally expressed by the quadratic assignment programming (QAP) form (Wang et al., 2019):

$$\underset{\mathbf{T} \in \{0,1\}^{N_1 \times N_2}}{\arg\min} \quad \text{vec}(\mathbf{T})^T \mathbf{K} \text{vec}(\mathbf{T}), \ s.t., \ \mathbf{T1} = \mathbf{1}, \ \mathbf{T}^T \mathbf{1} = \mathbf{1}, \tag{2}$$

where $\mathbf{T}$ is a binary permutation matrix encoding node correspondence, and $\mathbf{1}$ denotes a column vector with all elements to be one. $\mathbf{K} \in \mathbb{R}^{N_1 \times N_2}$ is the so-called affinity matrix (Leordeanu & Hebert, 2005), whose elements encode the node-to-node affinity between $\mathcal{G}_1$ and $\mathcal{G}_2$.

**Mutual Information.** Indeed, Eq. 1 is equivalent to maximizing the mutual information between $\mathcal{G}$ and $\mathcal{G}_S$. Namely, the goal of an explainer is to derive a small subgraph $\mathcal{G}_S$ such that:

$$\underset{\mathcal{G}_S \subset \mathcal{G}, |\mathcal{G}_S| \leq K}{\arg\max} \quad I(f_Y(\mathcal{G}_S); f_Y(\mathcal{G})), \tag{3}$$

where $I(\cdot; \cdot)$ refers to the Shannon mutual information between two random variables. Notably, instead of merely optimizing the information hidden in $\mathcal{G}_S$, another line of research (Yuan et al., 2021) seeks to reduce the mutual information between the subtracted subgraph $\mathcal{G} - \mathcal{G}_S$ and $\mathcal{G}$, *i.e.*,

$$\underset{\mathcal{G}_S \subset \mathcal{G}, |\mathcal{G}_S| \leq K}{\arg\min} \quad I(f_Y(\mathcal{G} - \mathcal{G}_S); f_Y(\mathcal{G})). \tag{4}$$

# 3    THE MATCHEXPLAINER APPROACH

In this section, we first introduce the formulation of our MatchExplainer approach from the perspective of mutual information maximization. We then divide the maximization objective into two parts, one addressed by a graph matching algorithm, and the other one tackled by subgraph ranking.

## 3.1    FORMULATION OF MATCHEXPLAINER

The core idea of MatchExplainer is to jointly optimize the two objectives in Eq. 3 and Eq. 4 as:

$$\underset{\mathcal{G}_S \subset \mathcal{G}, |\mathcal{G}_S| \leq K}{\arg\max} \quad I(f_Y(\mathcal{G}_S); f_Y(\mathcal{G})) - I(f_Y(\mathcal{G} - \mathcal{G}_S); f_Y(\mathcal{G})), \tag{5}$$

which is able to ensure the sufficiency (the first term) and necessity (the second term) of the replacement of $\mathcal{G}$ with its subgraph $\mathcal{G}_S$. However, this problem is nontrivial to solve owing to the exponential complexity of searching the desirable subgraph. Recent methods (Yuan et al., 2021; 2020; Wang et al., 2021b) employ another black-box GNN $f_S(\mathcal{G})$ to output $\mathcal{G}_S$. As mentioned before, such type of methods encounter the irrationality of pursuing explainability via unexplainable models.

Instead of solving Eq. 5 directly, we reduce it into a surrogate task with the aid of external graphs. Particularly, we fetch another graph $\mathcal{G}'$ that shares the same predicted property as $\mathcal{G}$ (*i.e.*, $\hat{y}_{\mathcal{G}} = \hat{y}_{\mathcal{G}'}$), and then extract the most relevant part between $\mathcal{G}$ and $\mathcal{G}'$ as the candidate $\mathcal{G}_S$. We will go through all possible $\mathcal{G}'$s, and locate the one giving the largest mutual information. In form, we derive:

$$\underset{\mathcal{G}' \in \mathbb{D}_{\mathcal{G}}, \mathcal{G}' \neq \mathcal{G}}{\arg\max} \left[ \underset{\mathcal{G}_S \subset \mathcal{G}, |\mathcal{G}_S| \leq K}{\arg\max} I(f_Y(\mathcal{G}_S); f_Y(\mathcal{G}'); f_Y(\mathcal{G})) - I(f_Y(\mathcal{G} - \mathcal{G}_S); f_Y(\mathcal{G})) \right], \tag{6}$$

where $\mathbb{D}_{\mathcal{G}} := \{\mathcal{G}' \mid y_{\mathcal{G}'} = y_{\mathcal{G}}\}$ denotes the set of graphs sharing the same label as $\mathcal{G}$. We define the optimal solution of Eq. 5 as $\mathcal{G}_S^*$, and immediately have the following proposition:

**Proposition 1** *If there exists $\mathcal{G}' \in \mathbb{D}_{\mathcal{G}}$ satisfying $f_Y(\mathcal{G}_S^*) = f_Y(\mathcal{G}')$, then Problem 5 is equivalent to Problem 6.*

We further divide Problem 6 into two subtasks:

- **Subgraph Matching**: Given any external graph $\mathcal{G}'$, we derive the most relevant subgraph $\mathcal{G}_S(\mathcal{G}')$ by $\mathcal{G}_S(G') = \arg\max_{\mathcal{G}_S \subset \mathcal{G}, |\mathcal{G}_S| \leq K} I(f_Y(\mathcal{G}_S); f_Y(\mathcal{G}'); f_Y(\mathcal{G}))$. Note that with the reference of $\mathcal{G}'$, it is efficient to derive $\mathcal{G}_S(\mathcal{G}')$ by applying a graph matching algorithm. We provide more details in § 3.2.

- **External Graph Ranking**: Different external graph $\mathcal{G}'$ corresponds to different subgraph $\mathcal{G}_S(G')$ returned by the last sub-problem. We choose the one that gives the smallest mutual information, that is $\mathcal{G}_S^* = \arg\min_{\mathcal{G}_S(\mathcal{G}')} I(f_Y(\mathcal{G} - \mathcal{G}_S(\mathcal{G}')); f_Y(\mathcal{G}))$. The details are given in § 3.3

Although modeling Problem 6 as the two subtasks above approximate the exact solution to some extent, we find in our experiments that they are sufficient to derive promising performance.

## 3.2 SUBGRAPH MATCHING MECHANISM

To begin with, we break the target GNN $h_Y$ into two consecutive parts: $h_Y = \phi_G \circ \phi_X$, where $\phi_G$ is the aggregator to compute the graph-level representation and predict the properties, and $\phi_X$ is the feature function to update both the node and edge features. For a given graph $\mathcal{G}$ with node features $\mathbf{h}_i \in \mathbb{R}^{\psi_v}, \forall i \in \mathcal{V}$ and edge features $\mathbf{e}_{ij} \in \mathbb{R}^{\psi_e}, \forall (i,j) \in \mathcal{E}$, the renewed output is calculated as $\{\mathbf{h}_i'\}_{i \in \mathcal{V}}, \{\mathbf{e}_{ij}'\}_{(i,j) \in \mathcal{E}} = \phi_X \left(\{\mathbf{h}_i\}_{i \in \mathcal{V}}, \{\mathbf{e}_{ij}\}_{(i,j) \in \mathcal{E}}\right)$, which is forwarded into $\phi_G$ afterwards.

Our target is to find subgraphs $\mathcal{G}_S \subset \mathcal{G}$ and $\mathcal{G}_S' \subset \mathcal{G}'$ both with $K$ nodes to maximize $I(f_Y(\mathcal{G}_S); f_Y(\mathcal{G}_S'))$. Here we utilize the node correspondence-based distance $d_G$ as a substitution for measuring the mutual information between $\mathcal{G}_S$ and $\mathcal{G}_S'$, which is minimized as follows:

$$\min_{\mathcal{G}_S \subset \mathcal{G}, \mathcal{G}_S' \subset \mathcal{G}'} d_G(\mathcal{G}_S, \mathcal{G}_S') = \min_{\mathcal{G}_S \subset \mathcal{G}, \mathcal{G}_S' \subset \mathcal{G}'} \left( \min_{\mathbf{T} \in \Pi(\mathcal{G}_S, \mathcal{G}_S')} \langle \mathbf{T}, \mathbf{D}^{\phi_X} \rangle \right), \tag{7}$$

where $\mathbf{D}^{\phi_X}$ is the matrix of all pairwise distances between node features of $\mathcal{G}_S$ and $\mathcal{G}_S'$. Its element is calculated as $\mathbf{D}_{ij}^{\phi_X} = d_X(\mathbf{h}_i', \mathbf{h}_j') \, \forall i \in \mathcal{V}, j \in \mathcal{V}'$, where $d_X$ is the standard vector space similarity such as the Euclidean distance and the Hamming distance. The inner optimization is conducted over $\Pi(.,.)$, which is the set of all matrices with prescribed margins defined as:

$$\Pi(\mathcal{G}_S, \mathcal{G}_S') = \left\{ \mathbf{T} \in \{0,1\}^{K \times K} \,|\, \mathbf{T1} = \mathbf{1}, \, \mathbf{T}^T \mathbf{1} = \mathbf{1} \right\}. \tag{8}$$

Due to the NP-hard nature of graph matching (Loiola et al., 2007), we adopt the greedy strategy to optimize $d_G(\mathcal{G}_S, \mathcal{G}_S')$ and attain the subgraph $\mathcal{G}_S$. It is worth noting that the greedy algorithm does not guarantee to reach the globally optimal solution (Bang-Jensen et al., 2004), but can yield locally optimal solutions in a reasonable amount of time. After that, we feed $\mathcal{G}_S$ into $h_Y$ and examine its importance. If $h_Y(\mathcal{G}_S) = h_Y(\mathcal{G})$, then $\mathcal{G}_S$ is regarded as the potential explanations. Otherwise, $\mathcal{G}_S$ is abandoned since it cannot recover the information required by $h_Y$ to predict $\mathcal{G}$.

It is worth noting that our excavation of explanations through subgraph matching is significantly different from most traditional graph matching methods. The majority of graph matching algorithms (Zanfir & Sminchisescu, 2018; Sarlin et al., 2020; Wang et al., 2020; 2021a) typically establish node correspondence from a whole graph $\mathcal{G}_1$ to another whole graph $\mathcal{G}_2$. However, we seek to construct partial node correspondence between the subgraph of $\mathcal{G}_1$ and the subgraph of $\mathcal{G}_2$. Besides, most current graph matching architectures (Zanfir & Sminchisescu, 2018; Li et al., 2019; Wang et al., 2020; Papakis et al., 2020; Liu et al., 2021a) are deep learning-based. They utilize a network to forecast the relationship between nodes or graphs, which has several flaws. For instance, the network needs tremendous computational resources to be trained. More importantly, its effectiveness is unreliable and may fail in certain circumstances if the network is not delicately designed. To overcome these limitations, we employ a non-parametric subgraph matching paradigm, which is totally training-free and fast to explore the most informatively joint substructure shared by any pair of input instances.

## 3.3 EXTERNAL GRAPH RANKING

Since our MatchExplainer is able to discover a variety of possible explanatory subgraphs via subgraph matching, how to screen out the most informative one becomes a critical issue. In this subsection, we introduce the external graph ranking mechanism to sort out the optimal explanation. Ideally, $\mathcal{G}'$ ought to share the exact same explanatory substructure with $\mathcal{G}$, i.e., $\max I(f_Y(\mathcal{G}_S); f_Y(\mathcal{G}_S'))$. Meanwhile, $\mathcal{G} - \mathcal{G}_S(\mathcal{G}')$ should be independent to the property of the input graph, i.e., $\min I(f_Y(\mathcal{G} - \mathcal{G}_S(\mathcal{G}')); f_Y(\mathcal{G}))$. Therefore, there are two distinct principles for selecting the counterpart graphs. The first line is to seek $\mathcal{G}'$ that has as close the explanatory subgraph as possible to $\mathcal{G}$. The second line is to ensure that $\mathcal{G} - \mathcal{G}_S$ maintains little information relevant to the original property $f_Y(\mathcal{G})$.

Nevertheless, without sufficient domain knowledge regarding which substructure is majorly responsible for the graph property, it would be impossible for us to manually select the counterpart graph $\mathcal{G}'$ that satisfies $\mathcal{G}_S \approx \mathcal{G}'_S$. As a remedy, the node correspondence-based distance $d_G(\mathcal{G}_S, \mathcal{G}'_S)$ can be treated as the indicator for whether this pair of graphs enjoy a similar explanatory substructure.

Though $d_G(\mathcal{G}_S, \mathcal{G}'_S)$ is a feasible criterion to filtrate the most informative substructure, a more efficient way is to immediately minimize the mutual information between $\mathcal{G} - \mathcal{G}_S$ and $\mathcal{G}$. This corresponds to decrease $I(f_Y(\mathcal{G} - \mathcal{G}_S); f_Y(\mathcal{G}))$ in Equ. 6. Towards this goal, we remove the extracted subgraph $\mathcal{G}_S$ from $\mathcal{G}$ and aspire to confuse GNNs' predictions on the remaining part $\mathcal{G} - \mathcal{G}_S$. Mathematically, the optimal $\mathcal{G}'$ maximizes the difference between the prediction of the whole graph and the prediction of the graph that is subtracted by $\mathcal{G}_S$. In other words, we wish to maximize:

$$\Delta_\mathcal{G}(\mathcal{G}', h_Y) = h_Y^{c^*}(\mathcal{G}) - h_Y^{c^*}(\mathcal{G} - \mathcal{G}_S(\mathcal{G}')), \tag{9}$$

where $c^*$ is the ground truth class of $\mathcal{G}$ and $\mathcal{G}_S$ is the substructure via subgraph matching with $\mathcal{G}'$.

Then given any graph $\mathcal{G}$ and a reference graph set $\mathbb{D}_\mathcal{R} = \{\mathcal{G}_1, ..., \mathcal{G}_n\}$, we acquire all possible subgraphs via matching $\mathcal{G}$ to available graphs in $\mathbb{D}_\mathcal{R}$. Notably, not all graphs in $\mathbb{D}_\mathcal{R}$ are qualified counterparts. There are several intuitive conditions that the counterpart graph $\mathcal{G}'$ has to satisfy. First, $\mathcal{G}$ and $\mathcal{G}'$ should belong to the same category predicted by $h_Y$. Besides, $\mathcal{G}'$ needs to have at least $K$ nodes. Otherwise, $G_S$ would be smaller than the given constrained size. After the pairwise subgraph matching, we calculate their corresponding $\Delta_\mathcal{G}(., h_Y)$ and pick up the one that leads to the largest $\Delta_\mathcal{G}(., h_Y)$ as the optimal counterpart graph.

**Effectiveness vs. efficiency.** The time-complexity is always an important topic to evaluate the practicability of explainers. For our MatchExplainer, the size of the reference set, *i.e.*, $|\mathbb{D}_\mathcal{R}|$, plays a vital role in determining the time cost. However, a limited number of counterpart graphs can also prohibit it from exploring better explanatory subgraphs. Thus, it is non-trivial to balance the effectiveness and efficiency of MatchExplainer by choosing an appropriate size of $\mathbb{D}_\mathcal{R}$.

## 4 THE MATCHDROP METHODOLOGY

**Preventing the false positive sampling.** Deep graph learning faces unique challenges such as feature data incompleteness, structural data sparsity, and over-smoothing. To address these issues, a growing number of data augmentation techniques (Hamilton et al., 2017; Rong et al., 2019) have been proposed in the graph domain and shown promising outcomes. Among them, the graph sampling and node dropping (Feng et al., 2020; Xu et al., 2021) are two commonly used mechanisms. However, most previous approaches are completely randomized, resulting in false positive sampling and injecting spurious information into the training process. For instance, *1,3-dinitrobenzene* ($C_6H_4N_2O_4$) is a mutagen molecule and its explanation is the $NO_2$ groups (Debnath et al., 1991). If any edge or node of the $NO_2$ group is accidentally dropped or destroyed, the mutagenicity property no longer exists. And it will misguide GNNs if the original label is assigned to this molecular graph after node or edge sampling.

To tackle this drawback, recall that our MatchExplainer offers a convenient way to discover the most essential part of a given graph. It is natural to keep this crucial portion unchanged and only drop nodes or edges in the remaining portion. Based on this idea, we propose a simple but effective method dubbed MatchDrop, which keeps the most informative part of graphs found by our MatchExplainer and alters the less informative part (see Figure 1).

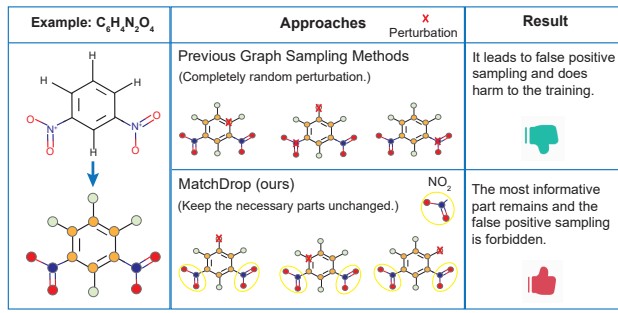

Figure 1: The illustration of our proposed MatchDrop.

The procedure of our MatchDrop is described as follows. To begin with, we train a GNN $h_Y$ for several epochs until it converges to an acceptable accuracy, which guarantees the effectiveness of the subsequent subgraph selection. Then

for each graph $\mathcal{G}$ in the training set $\mathbb{D}_{\text{train}}$, we randomly select another graph $\mathcal{G}' \in \mathbb{D}_{\text{train}}$ with the same class as the counterpart graph. Afterwards, we explore its subgraph $\mathcal{G}_S$ via MatchExplainer with a retaining ratio $\rho$ (*i.e.*, $|\mathcal{G}_S| = \rho|\mathcal{G}|$) and use it as the model input to train $h_Y$.

Notably, similar to the typical image augmentation skills such as rotation and flapping (Shorten & Khoshgoftaar, 2019), MatchDrop is a novel data augmentation technique for GNN training. However, instead of augmenting $\mathcal{G}$ randomly, MatchDrop reserves the most informative part and only changes the less important substructure. This significantly reduces the possibility of false positive sampling. Additionally, unlike other learnable mechanisms to inspect subgraphs, our MatchDrop is entirely parameter-free and, therefore, can be deployed at any stage of the training period.

**Training objective.** The training of GNNs is supervised by the cross entropy (CE) loss. Suppose there are $M$ classes in total, then the loss takes the following form:

$$\mathcal{L}_S = -\frac{1}{|\mathbb{D}_{\text{train}}|} \sum_{\mathcal{G} \in \mathbb{D}_{\text{train}}} \sum_{c=1}^{M} y_{\mathcal{G}} \log\left(h_Y^c\left(h_S(\mathcal{G}, \rho)\right)\right), \tag{10}$$

where $h_Y^c(.)$ indicates the predicted probability of $\mathcal{G}_S$ to be of class $c$ and $y_{\mathcal{G}}$ is the ground truth value. $h_S$ employs MatchExplainer to mine the subgraph $\mathcal{G}_S$ by matching $\mathcal{G}$ to a randomly selected counterpart graph $\mathcal{G}'$ in the training set $\mathbb{D}_{\text{train}}$ with a pre-defined ratio $\rho$.

## 5 EXPERIMENTAL ANALYSIS

### 5.1 DATASETS AND EXPERIMENTAL SETTINGS

Following Wang et al. (2021b), we use four standard datasets with various target GNNs.

- **Molecule graph classification**: MUTAG (Debnath et al., 1991; Kazius et al., 2005) is a molecular dataset for the graph classification problem. Each graph stands for a molecule with nodes for atoms and edges for bonds. The labels are determined by their mutagenic effect on a bacterium. The well-trained Graph Isomorphism Network (GIN) (Xu et al., 2018) has approximately achieved a 82% testing accuracy.

- **Motif graph classification.**: Wang et al. (2021b) create a synthetic dataset, BA-3Motif, with 3000 graphs. They take advantage of the Barabasi-Albert (BA) graphs as the base, and attach each base with one of three motifs: house, cycle, grid. We train an ASAP model (Ranjan et al., 2020) that realizes a 99.75% testing accuracy.

- **Handwriting graph classification**: Knyazev et al. (2019) transforms the MNIST images into 70K superpixel graphs with at most 75 nodes for each graph. The nodes are superpixels, and the edges are the spatial distances between them. There are 10 types of digits as the label. We adopt a Spline-based GNN (Fey et al., 2018) that gains around 98% accuracy in the testing set.

- **Scene graph classification**: Wang et al. (2021b) select 4443 pairs of images and scene graphs from Visual Genome (Krishna et al., 2017) to construct the VG-5 dataset (Pope et al., 2019). Each graph is labeled with five categories: stadium, street, farm, surfing and forest. The regions of objects are represented as nodes, while edges indicate the relationships between object nodes. We train an AAPNP (Klicpera et al., 2018) that reaches 61.9% testing accuracy.

We compare our MatchExplainer with several state-of-the-art and popular explanation baselines, which are listed below:

- **SA** (Baldassarre & Azizpour, 2019) directly uses the gradients of the model prediction with respect to the adjacency matrix of the input graph as the importance of edges.

- **Grad-CAM** (Selvaraju et al., 2017; Pope et al., 2019) uses the gradients of any target concept such as the motif in a graph flowing into the final convolutional layer to produce a coarse localization map highlighting the important regions in the graph for predicting the concept.

- **GNNExplainer** (Ying et al., 2019) optimizes soft masks for edges and node features to maximize the mutual information between the original predictions and new predictions.

Table 1: Comparisons of our MatchExplainer with other baseline explainers.

| | MUTAG | VG-5 | MNIST | BA-3Motif | |
| | ACC-AUC | ACC-AUC | ACC-AUC | ACC-AUC | Recall@ 5 |
|---|---|---|---|---|---|
| SA | 0.769 | 0.769 | 0.559 | 0.518 | 0.243 |
| Grad-CAM | $0.786 \pm 0.011$ | $0.909 \pm 0.005$ | $0.581 \pm 0.009$ | $0.533 \pm 0.003$ | $0.212 \pm 0.002$ |
| GNNExplainer | $0.895 \pm 0.010$ | $0.895 \pm 0.003$ | $0.535 \pm 0.013$ | $0.528 \pm 0.005$ | $0.157 \pm 0.002$ |
| PG-Explainer | $0.631 \pm 0.008$ | $0.790 \pm 0.004$ | $0.504 \pm 0.010$ | $\underline{0.586 \pm 0.004}$ | $0.293 \pm 0.001$ |
| PGM-Explainer | $0.714 \pm 0.007$ | $0.792 \pm 0.001$ | $0.615 \pm 0.003$ | $0.575 \pm 0.002$ | $0.250 \pm 0.000$ |
| ReFine | $\underline{0.955 \pm 0.005}$ | $\underline{0.914 \pm 0.001}$ | $\underline{0.636 \pm 0.003}$ | $0.576 \pm 0.013$[1] | $\underline{0.297 \pm 0.000}$[1] |
| MatchExplainer | **0.997** | **0.993** | **0.938** | **0.634** | **0.305** |
| Relative Impro. | 4.5% | 8.6% | 48.9% | 8.1% | 2.6% |

- **PGExplainer** (Luo et al., 2020) hires a parameterized model to decide whether an edge is important, which is trained over multiple explained instances with all edges.

- **PGM-Explainer** (Vu & Thai, 2020) collects the prediction change on the random node perturbations, and then learns a Bayesian network from these perturbation-prediction observations, so as to capture the dependencies among the nodes and the prediction.

- **Refiner** (Wang et al., 2021b) exploits the pre-training and fine-tuning idea to develop a multi-grained GNN explainer. It has both a global understanding of model workings and local insights on specific instances.

As the ground-truth explanations are usually unknown, it is tough to quantitatively evaluate the excellence of explanations. There, we follow Wang et al. (2021b) and employ **the predictive accuracy (ACC@$\rho$)** and **Recall@$N$** as the metrics. Specifically, ACC@$\rho$ measures the fidelity of the explanatory subgraphs by forwarding them into the target model and examining how well it recovers the target prediction. ACC-AUC is reported as the area under the ACC curve over different selection ratios $\rho \in \{0.1, 0.2, ..., 1.0\}$. Recall@$N$ is computed as $\mathbb{E}_{\mathcal{G}} [|\mathcal{G}_s \cap \mathcal{G}_S^*| / |\mathcal{G}_S^*|]$, where $\mathcal{G}_S^*$ is the ground-truth explanatory subgraph. Remarkably, Recall@$N$ is only suitable for BA3-motif, since this dataset is synthetic and the motifs are foregone.

## 5.2 CAN MATCHEXPLAINER FIND BETTER EXPLANATORY SUBGRAPHS?

### 5.2.1 QUANTITATIVE EVALUATIONS

To investigate the effectiveness of MatchExplainer, we conduct broad experiments on four datasets and the comparisons are reported in Table 1. For MUTAG, VG-5, and BA3-Motif, we exploit the whole training and validation data as the reference set. For MNIST, we randomly select 10% available samples as the reference set to speed up matching. It can be found that MatchExplainer outperforms every baseline in all cases. Particularly, previous explainers fail to explain GNNs well in MNIST with ACC-AUCs lower than 65%, but MatchExplainer can reach as high as 93.8%. And if we use the whole training and validation data in MNIST as the reference, its ACC-AUC can increase to 97.2%. This phenomenon demonstrates the advantage of subgraph matching in explaining GNNs when the dataset has clear patterns of explanatory subgraphs. Additionally, MatchExplainer also achieves significant relative improvements over the strongest baseline by 8.6% and 8.1% in VG-5 and BA3-Motif, respectively.

Furthermore, it is also worth noting that MatchExplainer realizes nearly 100% ACC-AUCs in each task but BA-3Motif. For BA-3Motif, we find that its predictive accuracy are $[0.31, 0.31, 0.31, 0.34, 0.49, 0.71, 0.97, 1.0, 1.0, 1.0]$ with different selection ratios. This aligns with the fact that most motifs in this task occupy a large fraction of the whole graph. Once the selection ratio is greater than 0.7, MatchExplainer is capable of figuring out the correct explanatory subgraph.

We visualize the explanations of MatchExplainer on MUTAG in Appendix B for qualitative evaluations.

---

[1]These results are reproduced

Table 2: Efficiency studies of different methods (in seconds).

| Method | Phase | MUTAG | VG-5 | MNIST | BA-3Motif |
|--------|-------|-------|------|-------|-----------|
| GNNexplainer | Training | 186.0 | 1127.2 | 1135.4 | 66.1 |
| | Inference (per graph) | 1.290 | 0.565 | 0.732 | 0.517 |
| | Training + Inference (total) | 703.4 | 1644.6 | 1782.1 | 271.6 |
| PG-Explainer | Training | 186.3 | 286.3 | 1154.1 | 112.4 |
| | Inference (per graph) | 0.056 | 0.094 | 0.025 | 0.020 |
| | Training + Inference (total) | 208.6 | 309.5 | 1162.1 | **120.4** |
| Refine | Training | 1191.6 | 1933.3 | 5025.8 | 763.0 |
| | Inference (per graph) | 0.068 | 0.107 | 0.026 | 0.027 |
| | Training + Inference (total) | 1218.9 | 1959.7 | 5051.2 | 773.8 |
| MatchExplainer | Training | – | – | – | – |
| | Inference (per graph) | 0.485 | 0.732 | 0.682 | 7.687 |
| | Training + Inference (total) | **194.6** | **180.3** | **667.8** | 3052.1 |

### 5.2.2 EFFICIENCY STUDIES

We compute the average inference time cost for each dataset with different methods to obtain explanations of a single graph. We also count the overall training and inference time expenditure, and summarize the results in Table 2. Specifically, we train GNNExplainer and PG-Explainer for 10 epochs, and pre-train Refine for 50 epochs before evaluation. It can be observed that though prior approaches enjoy fast inference speed, they suffer from long-term training phases. As an alternative, our MatchExplainer is completely training-free. When comparing the total time, MatchExplainer is the least computationally expensive in MUTAG, VG-5 and MNIST. However, as most motifs in BA-3Motif are large-size, MatchExplainer has to traverse a large reference set to obtain appropriate counterpart graphs, which unavoidably results in spending far more time.

## 5.3 CAN MATCHDROP GENERALLY IMPROVE THE PERFORMANCE OF GNNS?

### 5.3.1 IMPLEMENTATIONS

We take account of two backbones: GCN (Kipf & Welling, 2016), and GIN (Xu et al., 2018) with a depth of 6. Similar to Rong et al. (2019), we adopt a random hyper-parameter search for each architecture to enable more robust comparisons. There, *DropNode* stands for randomly sampling subgraphs, which can be also treated as a specific form of node dropping. *FPDrop* is the opposite operation of our MatchDrop, where the subgraph sampling or node dropping is only performed in the explanatory subgraphs while the rest remains the same. We add FPDrop as a baseline to help unravel the reason why MatchDrop works. *PGDrop* is similar to MatchDrop, but uses a fixed PGExplainer (Luo et al., 2020) to explore the informative substructure. The selection ratios $\rho$ for FPDrop, PGDrop, and MatchDrop are all set as 0.95.

### 5.3.2 OVERALL RESULTS

Table 3 documents the performance on all datasets except BA-3Motif, since its testing accuracy has already approached 100%. It can be observed that MatchDrop consistently promotes the testing accuracy for all cases. Exceptionally, FPdrop imposes a negative impact over the performance of GNNs. This indicates that false positive sampling does harm to the conventional graph augmentation methods, which can be surmounted by our MatchDrop effectively. On the other hand, PGDrop also gives rise to the decrease of accuracy. One possible reason is that parameterized explainers like PGExplainr are trained on samples that GNNs predict correctly, so they are incapable to explore explanatory subgraphs on unseen graphs that GNNs forecast mistakenly.

Table 3: Testing accuracy (%) comparisons on different backbones with and without MatchDrop.

| Dataset | Backbone | Original | FPDrop | DropNode | PGDrop | MatchDrop |
|---------|----------|----------|--------|----------|--------|-----------|
| MUTAG | GCN | $0.828 \pm 0.004$ | $0.803 \pm 0.017$ | $0.832 \pm 0.008$ | $0.825 \pm 0.02$ | **0.844±0.006** |
| | GIN | $0.832 \pm 0.003$ | $0.806 \pm 0.020$ | $\underline{0.835 \pm 0.009}$ | $0.828 \pm 0.01$ | **0.845±0.007** |
| VG-5 | GCN | $0.619 \pm 0.003$ | $0.587 \pm 0.014$ | $\underline{0.623 \pm 0.007}$ | $0.604 \pm 0.002$ | **0.638±0.008** |
| | GIN | $0.621 \pm 0.004$ | $0.593 \pm 0.018$ | $\underline{0.622 \pm 0.006}$ | $0.600 \pm 0.004$ | **0.630±0.003** |
| MNIST | GCN | $0.982 \pm 0.001$ | $0.955 \pm 0.008$ | $\underline{0.982 \pm 0.002}$ | $0.975 \pm 0.003$ | **0.986±0.002** |
| | GIN | $0.988 \pm 0.001$ | $0.959 \pm 0.005$ | $\underline{0.989 \pm 0.001}$ | $0.979 \pm 0.002$ | **0.990±0.001** |

## 6 RELATED WORK

### 6.1 EXPLAINABILITY OF GNNS

Though increasing interests have been appealed in explaining GNNs, the study in this area is still insufficient compared to the domain of images and natural languages. Generally, there are two research lines. The widely-adopted one is the parametric explanation methods. They run a parameterized model to dig out informative substructures, such as GNNExplainer (Ying et al., 2019), PGExplainer (Luo et al., 2020), and PGM-Explainer (Vu & Thai, 2020). The other line is the non-parametric explanation methods, which employ heuristics like gradient-like scores obtained by back-propagation as the feature contributions (Baldassarre & Azizpour, 2019; Pope et al., 2019; Schnake et al., 2020). Nevertheless, the latter usually shows much poorer results than the former parametric methods. In contrast, our MatchExplainer procures state-of-the-art performance astonishingly.

### 6.2 GRAPH AUGMENTATIONS

Data augmentation has recently attracted growing attention in graph representation learning to counter issues like data noise and data scarcity (Zhao et al., 2022). The related work can be roughly broken down into *feature-wise* (Zhang et al., 2017; Liu et al., 2021b; Taguchi et al., 2021), *structure-wise* (You et al., 2020; Zhao et al., 2021b), and *label-wise* (Verma et al., 2019) categories based on the augmentation modality (Ding et al., 2022). Among them, many efforts are raised to augment the graph structures. Compared with adding or deleting edges (Xu et al., 2022), the augmentation operations on node sets are more complicated. A typical application is to promote the propagation of the whole graph by inserting a supernode (Gilmer et al., 2017), while Zhao et al. (2021a) interpolate nodes to enrich the minority classes. On the contrary, some implement graph or subgraph sampling by dropping nodes for different purposes, such as scaling up GNNs (Hamilton et al., 2017), enabling contrastive learning (Qiu et al., 2020), and prohibiting over-fitting and over-smoothing (Rong et al., 2019). Nonetheless, few of those graph sampling or node dropping approaches manage to find augmented graph instances from the input graph that best preserve the original properties.

## 7 CONCLUSION

This paper proposes a subgraph matching technique called MatchExplainer for GNN explanations. Distinct from the popular trend of using a parameterized network that lacks interpretability, we design a non-parametric algorithm to search for the most informative joint subgraph between a pair of graphs. Furthermore, we combine MatchExplainer with the classic graph augmentation method and show its great capacity in ameliorating the false positive sampling challenge. Experiments convincingly demonstrate the efficacy of our MatchExplainer by winning over parametric approaches with significant margins.

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

## A  EXPERIMENTAL DETAILS

**Explaining GNNs.**  All experiments are conducted on a single A100 PCIE GPU (40GB). For the parametric methods containing GNNExplainer, PGExplainer, PGM-Explainer, and Refine, we use the reported performance in Wang et al. (2021b). Regarding the re-implementation of Refine in BA-3Motif, we use the original code with the same hyperparamters, and we adopt Adam optimizer (Kingma & Ba, 2014) and set the learning rate of pre-training and fine-tuning as 1e-3 and 1e-4, respectively.

**Graph augmentations.**  All experiments are also implemented on a single A100 PCIE GPU (40GB). We employ three sorts of different GNN variants (GCN, GAT, and GIN) to fit these datasets and verify the efficacy of various graph augmentation methods. We employ Adam optimizer for model training. For MUTAG, the batch size is 128 and the learning rate is 1e-3. For BA3-Motif, the batch size is 128 and the learning rate is 1e-3. For VG-5, the batch size is 256 and the learning rate is 0.5 * 1e-3. We fix the number of epochs to 100 for all datasets.

## B  EXPLANATIONS FOR GRAPH CLASSIFICATION MODELS

In this section, we report visualizations of explanations in Figure 2.

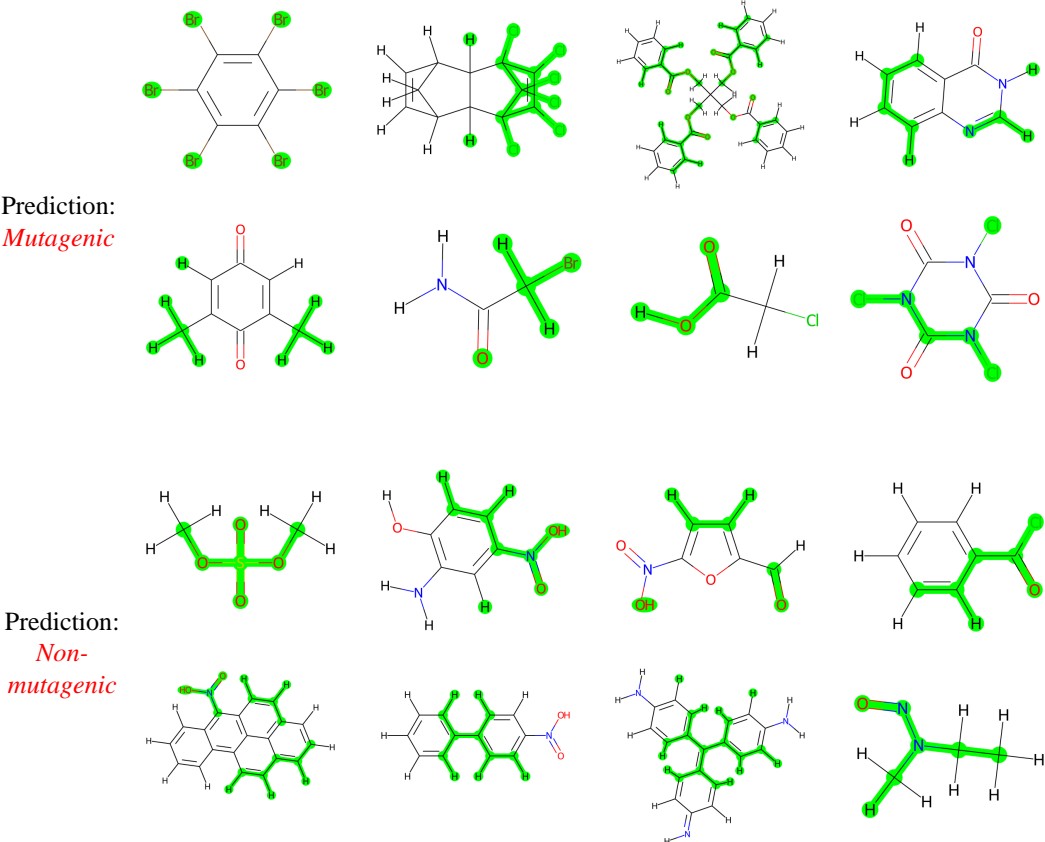

Figure 2: Explanatory subgraphs in Mutagenicity, where 50% nodes are highlighted.

