# OpenReview forum: "Rethinking the Explanation of Graph Neural Network via Non-parametric Subgraph Matching"
_ICLR.cc/2023/Conference — Submitted to ICLR 2023_

### Official Review · Reviewer_ncDW · 2022-10-24

**Confidence:** 5
**Clarity, Quality, Novelty And Reproducibility:** Overall, the main claims are not that…
**Correctness:** 3
**Technical Novelty And Significance:** 2
**Empirical Novelty And Significance:** 2
**Recommendation:** 3

**Strength And Weaknesses:**

Strength:
- The overall idea of generating explanations via exploring common structure from the graphs belonging to the same class is well-conceived.
- The proposed approach is well-designed and sounds.


Weaknesses:
The paper comes with a number of weaknesses as well. The most serious one is:
- The motivation is not convincing. This paper claims that the behavior of parametric explainers is hard to interpret since there are parametric. Thus depending on these black-box explainers to comprehend the prediction of the target black-box model is sub-optimal. It is the core motivation of this paper but I think: 1) parametric explainer is not hard to interpret. 2) even if the parametric explainer is hard to interpret, MatchExplainer still does not solve or remit this issue. To be more specific:
    - The procedure of the post-hoc explainers like GNNExplainer, PGExplainer, PGMExplainer, CF-GNNExplainer, GraphMask, SubgraphX, Gem, Refine, RCExplainer and GSAT (in its post-hoc working mode) have scrupulously theoretical support and is not hard to interpret. For example, if the important feature is masked or deleted by GNNExplainer or Gem, the outcome will fluctuate markedly. All the motivations and operations are accessible. I do not think the inherent issue claimed by the authors is existing.

- Note that MatchExplainer feeds the subgraph into the GNN model and selects the explanations based on the output.  Even if the parametric explainer is hard to interpret, the above operation inevitably brings MatchExplainer into the scope of “hard to interpret”, since taking $I(f(G_s), f(G))$ as the proxy of $I(G_s,G)$ introduces the black-box model $f$. That is, MatchExplainer is also a black box because it can not explain why $I(G_s,G)= I(f(G_s), f(G))$.

In a nutshell, if this paper only generates explanations via a more efficiently common subgraph/structure searching algorithm from the graphs belonging to one class, it really has a reasonable and novel motivation.

Other less critical, but still major weaknesses are:
- The paper is poorly structured. The part of “SUBGRAPH MATCHING MECHANISM” is wordy and the consistency of the part 3 and part 4 is poor. In a nutshell, I suggest that the paper should pay more attention to the part 3.2, not the part 4.
- Another criticism of the paper is its writing. For example, it has GNNExplainer in Table 2 but GNNexplainer in Table 2.


**Summary Of The Paper:**

This paper proposes to mine the explanatory motif in a subgraph-matching manner. Specifically, it investigates the explainability of GNNs from the perspective of non-parametric subgraph matching and proposes a novel method dubbed MatchExplainer. Moreover, it takes the advantage of MatchExplainer and introduces a simple technique called MatchDrop to enhance the traditional graph augmentation methods.

**Summary Of The Review:**

Essentially, I think there is some promising work. However, I think it really needs a more reasonable motivation, better presentation, and richer analysis to be accepted for publication at a top-tier conference. Therefore, I am leaning on the negative side.

---

### Official Review · Reviewer_mQZh · 2022-10-24

**Confidence:** 5
**Correctness:** 4
**Technical Novelty And Significance:** 4
**Empirical Novelty And Significance:** 4
**Recommendation:** 8

**Clarity, Quality, Novelty And Reproducibility:**

* The paper is clear in general. Perhaps the issues concerning information theoretic measures deserve more attention. For instance, I assume that Shannon entropy (Shannon MI) is used for one-dimensional labels. For multi-labeled instances, I suggest using any bypassing entropy estimator (e.g. Rényi ones).
* Novelty. Additionally, the point of data augmentation is very novel.
* Reproducibility: as usual, no code was released (even anonymized). Should be done, in my opinion. Sometimes implementation details can be discovered in the code (for instance how was the explainability quantified)

**Details Of Ethics Concerns:**

OK

**Strength And Weaknesses:**

* Strengths. The idea of extracting informative subgraph is very interesting. Although this implies approximating a NP-hard problem, the experimental results show that greedy graph-subgraph matching is good enough.
* Weaknesses: The protocol for quantifying explainability could be better clarified, but in general one can follow the paper.

**Summary Of The Paper:**

Main concern. Transform GCN explainability into subgraph-graph matching.

Protocol for quantifying the explainability: “As the ground-truth explanations are usually unknown, it is tough to quantitatively evaluate the excellence of explanations. There, we follow Wang et al. (2021b) and employ the predictive accu- racy (ACC@ρ) and Recall@N as the metrics “.
It is clear how to apply the protocol to the proposed method (e.g. get the reported explainable subgraph and measure its accuracy). However, it is not clarified how the data in Table 1 is obtained for the baselines (my intuition is that you get the most explained edges and use them to check the accuracy, is that true?). This point is a bit clear for GNNEexplainer: “  They run a parameterized model to dig out informative substructures, such as GNNExplaine “. And for PG-Explainer. Please clarify the rest of the baselines.

Recommendation. Please explore efficient graph matchers such as the “bypass graph matching - Entropic Alignment (SIAM Imaging Science)” as well as it associated mutual information between graphs (Pattern Recognition Letters). In this latter approach, an structural (spectral embedding is used to transform graph matching into point alignment).


**Summary Of The Review:**

This is a nice paper in general, which contributes to a subgraph-graph method for elucidating explainability. The idea is fine since it suggests what parts of the structure are superfluous. As a result, the resulting substructure is more efficient. The results are fine though sometimes not clearly interpretable.

---

### Official Review · Reviewer_aTNz · 2022-10-24

**Confidence:** 4
**Correctness:** 2
**Technical Novelty And Significance:** 2
**Empirical Novelty And Significance:** 3
**Recommendation:** 3

**Clarity, Quality, Novelty And Reproducibility:**

The writing can be much improved. Many details about the methodology are missing in the current manuscript.

**Strength And Weaknesses:**

Strengths:
- I like the idea of not using the black-box explainers.
- The proposed method outperforms the baseline methods as shown by the experimental results.

Concerns/Questions:
My general concern is that the work is heavily relying on mutual information while it is lack of rigorous justification. Please see my detailed concerns/questions below:
- How is the equivalence between Eq. (1) and Eq. (3) established? My guess is that they are only equivalent under specific definition of risk function $\mathcal{R}$.
- For proposition 1, it would be nice if the authors could provide a rigorous proof (or at least proof sketch in the main body). I cannot find it either in the main body or the appendix.
- In section 3.2, can we just simply disentangle GNN into two disjoint parts? In many GNNs (e.g., GCN, GAT), graph structure is an important part in learning node/edge features (i.e., $\phi_X$ here) so it is hard to simply distangle them into the composition of $\phi_G$ and $\phi_X$. The authors may provide a few instantiations of $\phi_G$ and $\phi_X$ to avoid confusion.
- Why is the node correspondance-based distance $d_G$ a good substitution of the multivariate mutual information?
- Subgraph matching is not a new problem (see some references below). Some discussion on existing works of subgraph matching is needed.
[1] Mongiovi, Misael, et al. "Sigma: a set-cover-based inexact graph matching algorithm." Journal of bioinformatics and computational biology 8.02 (2010): 199-218.
[2] Du, Boxin, et al. "First: Fast interactive attributed subgraph matching." Proceedings of the 23rd ACM SIGKDD international conference on knowledge discovery and data mining. 2017.
[3] Kopylov, Alexei, and Jiejun Xu. "Filtering strategies for inexact subgraph matching on noisy multiplex networks." 2019 IEEE international conference on big data (big data). IEEE, 2019.
[4] Moorman, Jacob D., et al. "Subgraph matching on multiplex networks." IEEE Transactions on Network Science and Engineering 8.2 (2021): 1367-1384.
- Both section 3.2 and section 3.3 are a bit vague on the detailed methodology. More details on the key steps of each phase (matching and ranking) are expected. As a reader, I can only get the general idea but not knowing how I it works and how I should use the method.
- There should be a section to discuss (sub)graph matching in related work.
- The authors can better use the Appendix to thoroughly discuss the technical details, e.g., detailed description of key steps, pseudocode, theoretical justifications. Also, for Appendix B, it is better to have insightful analysis to showcase the efficacy of the proposed method.

**Summary Of The Paper:**

This paper studies the explainability of graph neural networks (GNNs) using subgraph matching. The paper first identifies the limitation of most existing works, which is the usage of black-box explainer to generate explanatory subgraph. Based on that, the authors propose a matching-then-ranking framework named MatchExplainer to find the explanatory subgraph. Experimental results demonstrate the efficacy of the proposed method.

**Summary Of The Review:**

The idea is interesting, but the writing is not satisfying. More rigorous justifications on its math are needed. The authors are expected to provide more details about the proposed method to avoid confusion.

---

### Official Review · Reviewer_Xwid · 2022-11-02

**Confidence:** 3
**Correctness:** 3
**Technical Novelty And Significance:** 2
**Empirical Novelty And Significance:** 2
**Recommendation:** 3

**Clarity, Quality, Novelty And Reproducibility:**

### Clarity
The paper is not very well written. However, the idea is clear. Notations can be toned down a bit. It seems simple explanations following the equations would be much helpful in improving the readability of the paper.

### Novelty
The proposed method does not seem to be high in novelty. The main part which should be of focus is the subgraph matching and how it can be done efficiently as well as effectively. The authors do not pay much attention to that. The rest of the approach seems to be set of simple heuristics.

**Strength And Weaknesses:**

### Strengths
1. The motivation of the paper is clear and the problem being solved is very relevant.
1. The addition of MatchDrop strengthens the paper with possible use for applications other than GNN explainability.
1. Exploring non-parametric methods for GNN explainability is interesting.

### Weaknesses
1. Firstly, the authors mention that it is irrational to use deep learning based method to explain GNN predictions. I disagree with this. The explainability problem is mathematically formulated (as well in this paper) just needs to get the subgraph which is sufficient to predict the output. Any method, whether learning based or non-parametric, if able to solve this, that is a rational method to solve the problem. We don’t need the explaining method to be explainable itself.
1. Rather than an algorithm, the method seems to be like a heuristic without much novelty. The main issue with non-learning based strategies for finding subgraphs is the complexity which is why learning based methods are promising.
1. Furthermore, the greedy strategy used to find the subgraph is not well discussed. It is important since that is the part of finding the subgraph. How effective and efficient the method is needs to be discussed.
1. Selecting subgraphs based only on node-correspondence distance is not properly motivated. Why should this give the best (or good enough) subgraph is not intuitive except that it may be easy to compute. Furthermore, even though there may be node similarities, it does not necessarily mean the corresponding subgraphs match. For example, two non-connected nodes may have high node-correspondence score but since they are not connected, it may not be a substructure which explains the prediction. This needs proper elaboration.
1. The method is restricted to finding subgraphs to size K. What value of K is used and experiments ablating this value are not discussed and shown. How imp is the value of K is not clear.
1. Why do we need to use the reference graph set? How do you decide the set of subgraphs in the reference set? Does it depend on the specific dataset used or can be general?
1. The time complexity is also not properly studied. What exactly is the complexity of MatchExplainer in Big-O notation and how does it relate to other models is not properly discussed.
1. Also in Table 2, training time is considered which is not very relevant. In terms of efficiency, we are mainly concerned with inference time. This is because, training is only one time whereas inference is relevant in deployment. Inference time is considerably higher compare to Refine and PG-Explainer.


**Summary Of The Paper:**

The paper deals with explainability of the predictions made by Graph Neural Networks. Most methods use a learning based approach to generate explanations for the GNN predictions. In this paper, the authors propose MatchExplainer, a non-parametric technique of finding the subgraph which best explains the predictions. In other words the subgraph which contains maximum information relevant to the target. For this, they use graph matching method of finding the common subgraphs between graphs on which GNNs give the same predictions. Furthermore, they use this subgraph to improve the graph augmentation techniques called MatchDrop. Experiments show that the proposed technique is useful and outperforms other deep learning based baselines on most datasets.


**Summary Of The Review:**

Overall, exploring non-parametric methods for explainability of GNNs is interesting. However, the authors doesn’t seem to be proposing an efficient way of finding subgraphs between graphs with same labels. This is precisely the reason, why other methods use deep-learning based techniques.

---

### Decision · Program_Chairs · 2023-01-20

**Decision:**

Reject

**Justification For Why Not Higher Score:**

Clear agreement on the paper lacking clarity and rigor. It is not ready for publication.

**Justification For Why Not Lower Score:**

N/A

**Metareview: Summary, Strengths And Weaknesses:**

Three of the four reviewers provided good reasons why the paper should be rejected. For instance, a lack of clarity in writing and a lack of rigor in the formal statements and their proofs.